# NiH-catalyzed anti-Markovnikov hydroamidation of unactivated alkenes with 1,4,2-dioxazol-5-ones for the direct synthesis of N-alkyl amides

Bingnan Du[1], Chun-Ming Chan[1], Yuxin Ouyang[1], Kalok Chan[2], Zhenyang Lin [2✉] & Wing-Yiu Yu [1✉]

The addition of a nitrogen-based functional group to alkenes via a direct catalytic method is an attractive way of synthesizing value-added amides. The regioselective hydroamidation of unactivated alkenes is considered one of the easiest ways to achieve this goal. Herein, we report the NiH-catalyzed anti-Markovnikov intermolecular hydroamidation of unactivated alkenes enabled by using 2,9-dibutylphenathroline (diBuphen) as the ligand. This protocol provides a platform for the direct synthesis of over 90 structurally diverse N-alkyl amides using dioxazolones, which can be easily derived from abundant carboxylic acid feedstocks. This method succeeds for both terminal and internal unactivated alkenes and some natural products. Mechanistic studies including DFT calculations reveal an initial reversible insertion/ elimination of the [NiH] to the alkene, followed by the irreversible amidation to furnish the N-alkyl amides. By crossover experiments and deuterium labeling studies, the observed anti-Markovnikov regioselectivities are suggested to be controlled by the sterical environment of the coupling reaction.

[1] State Key Laboratory of Chemical Biology and Drug Discovery, Department of Applied Biology and Chemical Technology, The Hong Kong Polytechnic University, Hung Hom, Kowloon, Hong Kong, PR China. [2] Department of Chemistry, The Hong Kong University of Science and Technology, Hong Kong, PR China. ✉email: chzlin@ust.hk; wing-yiu.yu@polyu.edu.hk

Amides are one of the most prevalent structures found in organic molecules, such as peptides, proteins, and DNA. It also constitutes an important motif in many functional materials and medicinal products[1–5]. While the conventional amide synthesis is performed by coupling carboxylic acids with amines, the production of stoichiometric amount unwanted side-products remains an environmental concern[6]. Recently, metal hydride-mediated hydrometalation using electrophilic aminating reagents received considerable attention for regioselective amines/amides synthesis (Metal = Fe[7,8], Co[9,10], Ni[11,12], Cu[13–15]). Pioneered by Buchwald and Miura, CuH-catalyzed regioselective hydroamination of alkenes with hydroxy amine derivatives was achieved to give the amines/amide products. The reaction was suggested to proceed through some *n*-alkyl-Cu(I) complexes[16,17]. CuH-catalyzed hydroaminations of styrene usually show Markovnikov selectivity due to the formation of a more stable benzyl-Cu complex. In contrast, the analogous hydrometalation of aliphatic alkenes often exhibits anti-Markovnikov regioselectivity due to steric constraints (Fig. 1a, up)[17]. Another interesting example was reported by Hartwig et al. that an $OBzCl_3$ group on alkenes would serve as a polarity-directing group for regioselective hydrometalation (Fig. 1a, down)[18].

Recently catalytic C–N bond formation was also enabled by the NiH-mediated hydroamidation of alkenes[11,12,19–22]. NiH hydrometalation of aliphatic alkenes is usually non-regioselective, giving both Markovnikov and anti-Markovnikov products. The regiocontrol of the NiH-mediated hydroamidation is further complicated by the propensity of the chain-walk isomerization via alkyl-Ni(II) species[23]. To achieve the proximal hydroamidation of alkenes, directing group can be used to suppressed the chain-walk isomerization, and the thermodynamic preferred five-membered nickelacycle further support the regioselective hydrometalation of some unactivated internal alkenes (Fig. 1b, up). Hong's group utilized an aminoquinoline as a directing group to mediate the regioselective hydrometalation of some unactivated internal alkenes[24]. Yet, this strategy is limited by the necessity of a pre-installed functional group nearby the C=C bond, and thus, hydroamidation to access linear amides are spare. Direct regioselective hydroamination of directing group-free alkenes may be appealing to the synthetic industry. During the preparation of this manuscript, our group also reported an NiH-catalyzed remote C(sp3)−H hydroamidation of alkenes[25]. Also, in this period, Chang et al. demonstrated the NiH-catalyzed hydroamidation of alkynes with dioxazolones to give enamides in (E)-anti-Markonikov or Markonikov selectivity. The regioselectivities are controlled by the choice of ligands, which govern the initial hydrometalation step (Fig.1b, down)[26].

Here we report the NiH-catalyzed anti-Markovnikov hydroamidation of alkenes with dioxazolones for the synthesis of structurally diverse linear amides (Fig. 1c). By employing 2,9-dibutyl-1,10-phenanthroline (diBuphen) as ligand, over 90 anti-Markovnikov N-alkyl amides were afforded in up to 92% yield. Mechanistic investigations combined with density functional theory (DFT) calculation suggested the observed selectivity can be understood by invoking Curtin-Hammett type equilibrium.

## Results
In this work, treating vinylcyclohexane **1** (0.2 mmol), 3-cyclobutyl-1,4,2-dioxazol-5-one **2** (0.4 mmol), pinacolborane (2.0 equiv) using $[Ni(ClO_4)_2]·6H_2O$ (10 mol %) with diBuphen (12 mol %) in THF (2.0 ml) at room temperature for 12 h furnished the anti-Markovnikov hydroamidation product **3** in 95% yield. Notably, <5% Markovnikov hydroamidation product **4** was detected (Table 1, entry 1). Other nickel precursors, such as $NiI_2$ and Ni(COD)$_2$, are found to be ineffective catalysts (entries 2–3).

The effect of some common organic solvents was then tested (entries 4–6). The reaction with DCE gave a better result (**3**: 90%), while running the reaction in MeCN and in DMA afforded **3** in 55% and 56% yields, respectively. To our delight, excellent regioselectivity and yield were achieved by employing a DMA/THF (1:9 v/v) mixture as the solvent system (entry 7); the desired **3** was obtained in 99% without any Markovnikov products formation. The effect of ligand was then studied, no desired products were obtained in the absence of the diBuphen ligand (entry 8). Employing related ligands such as **L1** led to unselective hydroamidation: (**3**: 9% and **4**: 10%) (entry 9). While **L2** was found to be less effective, affording a moderate hydroamidation yield with poor regioselectivity: (**3**: 26% and **4**: 35%) (entry 10). Reactions with **L3** as ligand produced comparable results as diBuphen: **3** (75%; entry 11). Diphosphine ligands such as **L5** and **L6** were ineffective for this hydroamidation reaction (entries 12–13). Apparently, the presence of HBpin is critical for this transformation; no desired products were formed without HBpin (entry 14). Other common hydride sources such as $Ph_2SiH_2$ and $(EtO)_2SiMeH$ were found to be ineffective (entries15–16).

With the protocol of anti-Markovnikov alkene hydroamidation established, we studied the scope of dioxazolones (Fig. 2). Under the Ni-catalyzed conditions, dioxazolones containing primary alkyl groups were effectively coupled to the vinylcyclohexane **1** to form the corresponding N-alkyl amides (**5–11**) in good yields (~80%). Dioxazolones containing ethoxy, phthalimide, thiophene, ketone and indole groups were converted to their amides (**12–16**) in good yields. Moreover, the coupling reaction of **1** with dioxazolones derived from 2-chloropropanoic acid was also successful to give **17** in 71% yield. Similarly, the dioxazolones containing secondary alkyl groups were transformed to the corresponding linear N-alkyl amides (**3**, **18–28**) in good yields (~80%). Presumably, due to steric reasons, transformations of the dioxazolones bearing tertiary alkyl groups were less successful (**29**: 25% and **30**: 23%). The reactions of the dioxazolones derived from 3-(methoxycarbonyl)bicyclo[1.1.1]pentane-1-carboxylic acid and methyl 4-(5-oxo-1,4,2-dioxazol-3-yl)cubane1-carboxylate containing strained carbocycles afforded **31** and **32** in 92% and 70% yields, respectively.

As anticipated, facile couplings with the dioxazolones containing aryl groups were also achieved, and **33–36** were formed in ~80% yields. For instance, dioxazolones derived from 6-fluoropicolinic acid and ferrocenecarboxylic acid produced **37** (76%) and **38** (47%). The analogous reactions with dioxazolones derived from some natural products gave the corresponding amides effectively, including 1-pyrenebutyric acid (**39**: 91%), Isoxepac (**40**: 63%), Citronellic acid (**41**: 78%) and Indomethacine (**42**: 40%). The molecular structure of **40** has been confirmed by X-ray crystallography. Less effective reactions were encountered for the dioxazolones bearing secondary or tertiary alkyl groups, such as Ibuprofen (**43**: 25%), Naproxen (**44**: 36%) and Gemfibrozil (**45**: 8%).

The synthetic versatility of this reaction was further explored with the alkenes scope (Fig. 3). With 3-cyclobutyl-1,4,2-dioxazol-5-one **2** as the nitrene source, we examined the reactivity of some terminal alkenes (**46–52**), and the corresponding amides were obtained in good yields (~80%). However, reaction of 3,3-dimethylbut-1-ene afforded the desired product **49** only in 33% yield. The relatively low yield could be attributed to steric interaction between the alkene and the Ni catalyst. Substrates with two C=C motifs, such as 4-vinylcyclohex-1-ene would undergo hydroamidation at the least hindered position, and **53** was formed in 47% yield. Hydroamidation of alkynes was also accomplished, and **54** was produced in 26% yield with hex-1-yne as substrate. Coupling reactions of 1,1-disubsituted alkenes gave **55–58** in moderate yields (~60%), indicative of the steric sensitivity of this reaction.

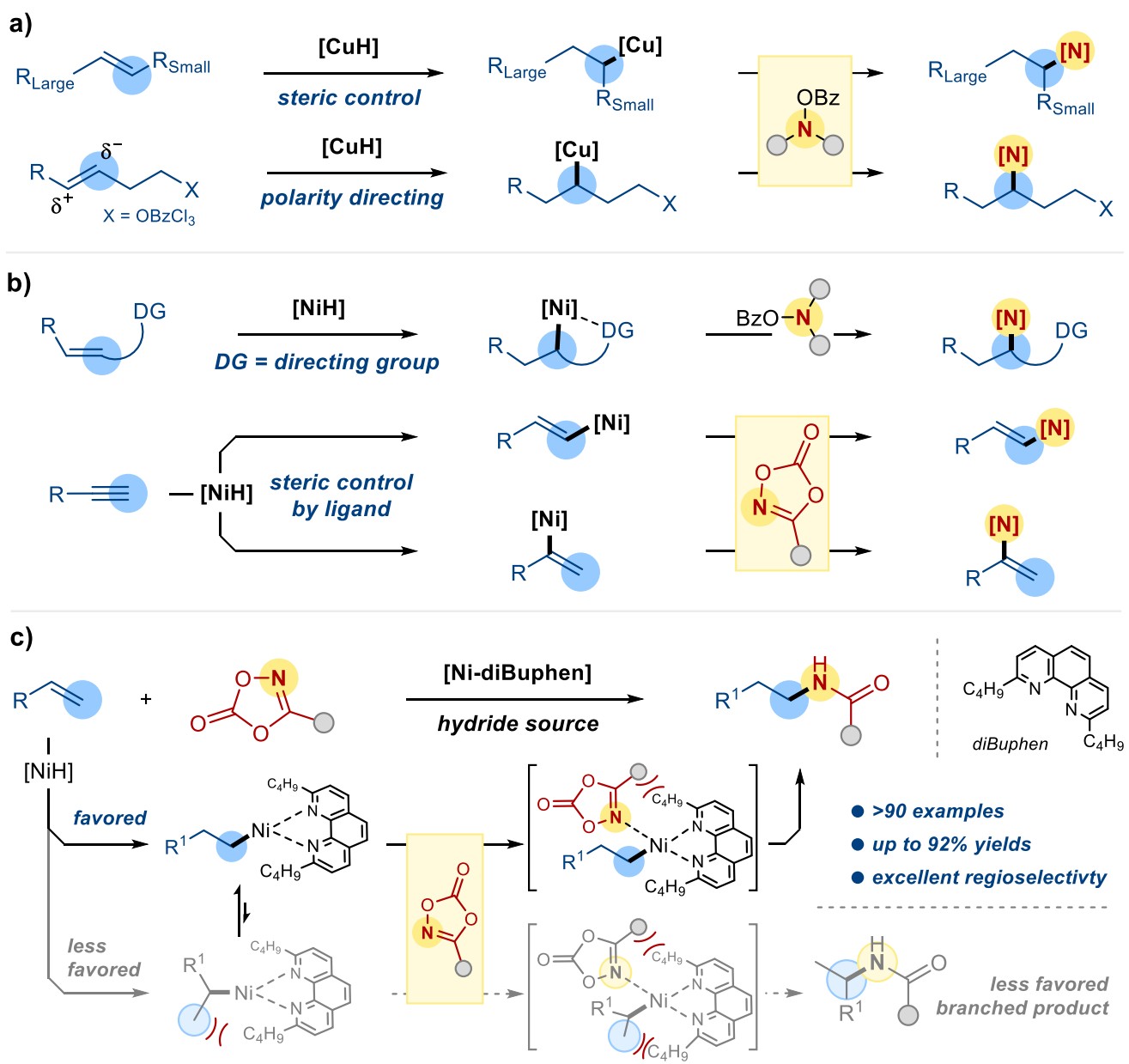

**Fig. 1 Strategies for [M−H]-catalyzed regioselective hydroamidation reactions. a** [CuH] mediated regioselective hydroamination. **b** [NiH] mediated regioselective hydroamination/hydroamidation. **c** This work: Ligand enhanced regioselective hydroamidation.

For internal alkenes, effective coupling reactions of **2** with cyclohexene and bicyclo[2.2.1]hept-2-ene were achieved to furnish **59** (49%) and **60** (62%), respectively. Facile transformation with cholesterol as substrate furnished **61** in 43% yield. We then turned to expand the alkene scope using a bulkier 3-(heptan-4-yl)-1,4,2-dioxol-5-one **62** for the reaction. Similar to our earlier examples, the hydroamidation of simple terminal alkenes produced **63–65** in good yields. Notably, the regioselectivity was reversed to branched amide products when but-3-enenitrile and allyltrimethylsilane were used. The desired linear products **66** and **67** were only obtained in 18 and 24% yields. Likewise, 2-allylcyclohexan-1-one reacted with **62** afforded the anti-Markovnikov product **68** (30%) accompanied with the branched products (50%). For the reaction with 1-allyl-*N,N*-diethylcyclopropane-1-sulfonamide, anti-Markovnikov product **69** was obtained exclusively in 74% yield.

Terminal alkenes with a four-carbon alkyl chain were also tested. The reactions of the alkenes with ester, phthalimide and epoxy substituents at the δ carbon displayed excellent selectivity for the formation of linear amides **70–72** albeit in moderate yields. However, ketone and amide substituted alkenes reacted with **62** afforded branched amides in 53% and 51% yields with the desired **73** and **74** being formed in 36 and 37%. For the terminal alkenes bearing a five-carbon alkyl chain such as 5-bromopent-1-ene, amide **75** was obtained in 44% yield. The reaction of hex-5-enenitrile gave the linear amide (**76**: 91%) in anti-Markovnikov selectivity; however, the analogous transformation of but-3-enenitrile only gave the anti-Markovnikov product **66** in 18% yield. Other alkenes with functional groups such as nitro, esters and alcohol derived motifs were effectively coupled to **62 to** afford the corresponding linear amides **77–83** in 59–79% yields. The reactivities of terminal alkenes bearing a six-carbon alkyl chain were

**Table 1 Reaction optimization[a].**

| Entry | Variation from initial conditions | 3 (%)[b] | 4 (%)[b] | rr[c] |
|---|---|---|---|---|
| 1 | None | 95 | <5 | >20:1 |
| 2 | NiI$_2$ as catalyst | 17 | <5 | 8:1 |
| 3 | Ni(COD)$_2$ as catalyst | <5 | <5 | 1:1 |
| 4 | DCE as solvent | 90 | <5 | >20:1 |
| 5 | MeCN as solvent | 55 | <5 | 15:1 |
| 6 | DMA as solvent | 56 | <5 | 15:1 |
| 7 | **DMA/THF (1:9 v/v) as solvent** | **99 (88)[d]** | **n.d.** | **–** |
| 8 | Without ligand | n.d. | n.d. | – |
| 9 | L1 instead of L4 | 9 | 10 | 1:1 |
| 10 | L2 instead of L4 | 26 | 35 | 1:1.5 |
| 11 | L3 instead of L4 | 75 | <5 | 20:1 |
| 12 | L5 instead of L4 | <5 | <5 | 1:1 |
| 13 | L6 instead of L4 | <5 | <5 | 1:1 |
| 14 | Without HBpin | n.d. | n.d. | – |
| 15 | Ph$_2$SiH$_2$ instead of HBpin | <5 | <5 | 1:1 |
| 16 | (EtO)$_2$SiMeH instead of HBpin | <5 | <5 | 1:1 |

The bold text highlights the optimized conditions.
[a]Reaction conditions: **1** (0.2 mmol), **2** (0.4 mmol), catalyst (10 mol %), ligand (12 mol %), hydride reagents (2.0 equiv) and solvent (2.0 ml) in N$_2$ at room temperature for 12 h unless otherwise specified.
[b]NMR yield.
[c]Regioisomeric ratio (**rr**) is defined as linear/branched.
[d]Isolated yield.

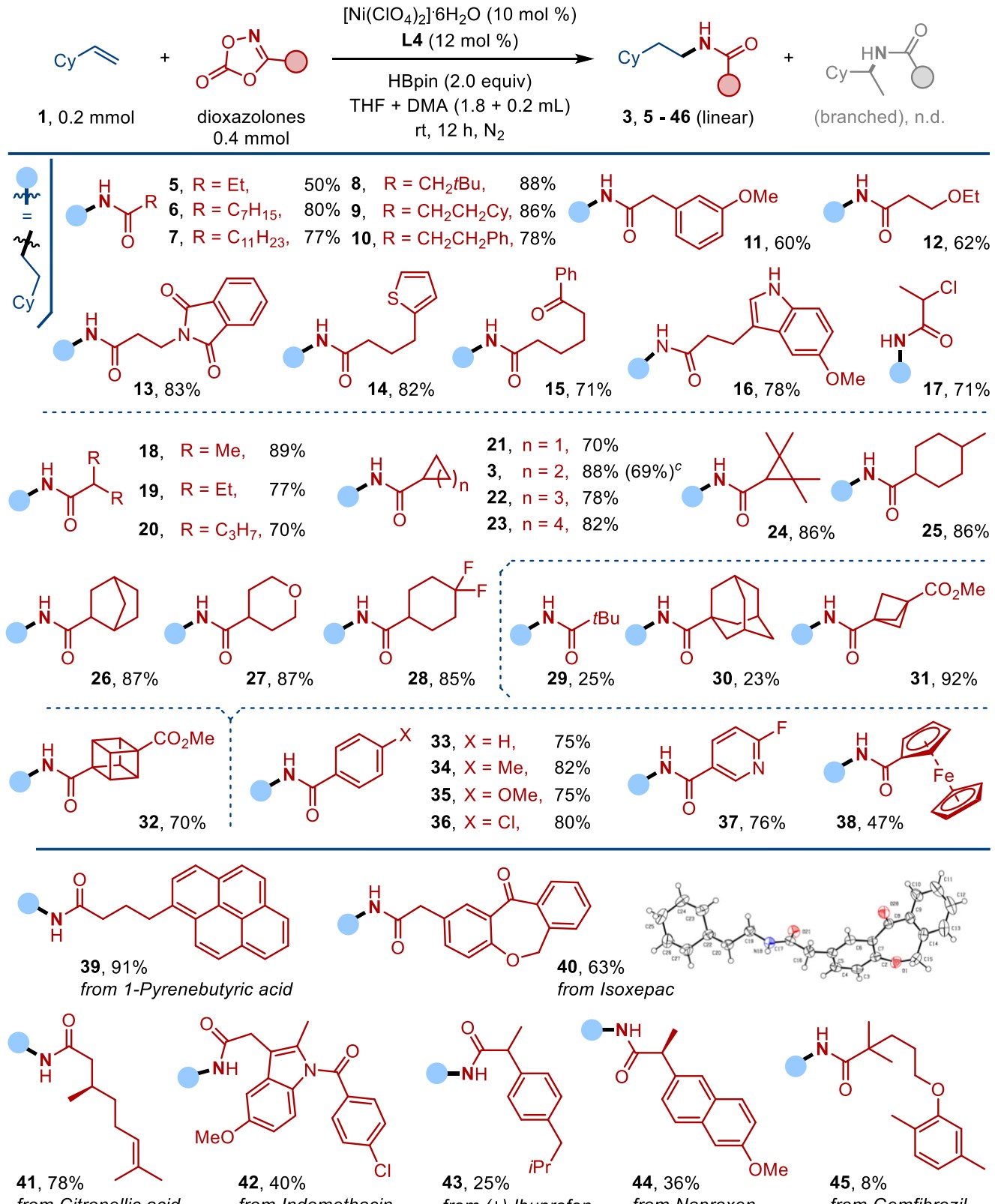

**Fig. 2 Scope of dioxazolones.** [a]Reaction conditions: **1** (0.2 mmol), dioxazolones (0.4 mmol), [Ni(ClO$_4$)$_2$]•6H$_2$O (10 mol %), **L4** (12 mol %), HBpin (2.0 equiv), THF + DMA (1.8 + 0.2 ml) in N$_2$ at room temperature for 12 h unless otherwise specified. [b]Isolated yield. [c]Large scale reaction: **1** (10.0 mmol) was used, **3** was obtained in 1.45 g.

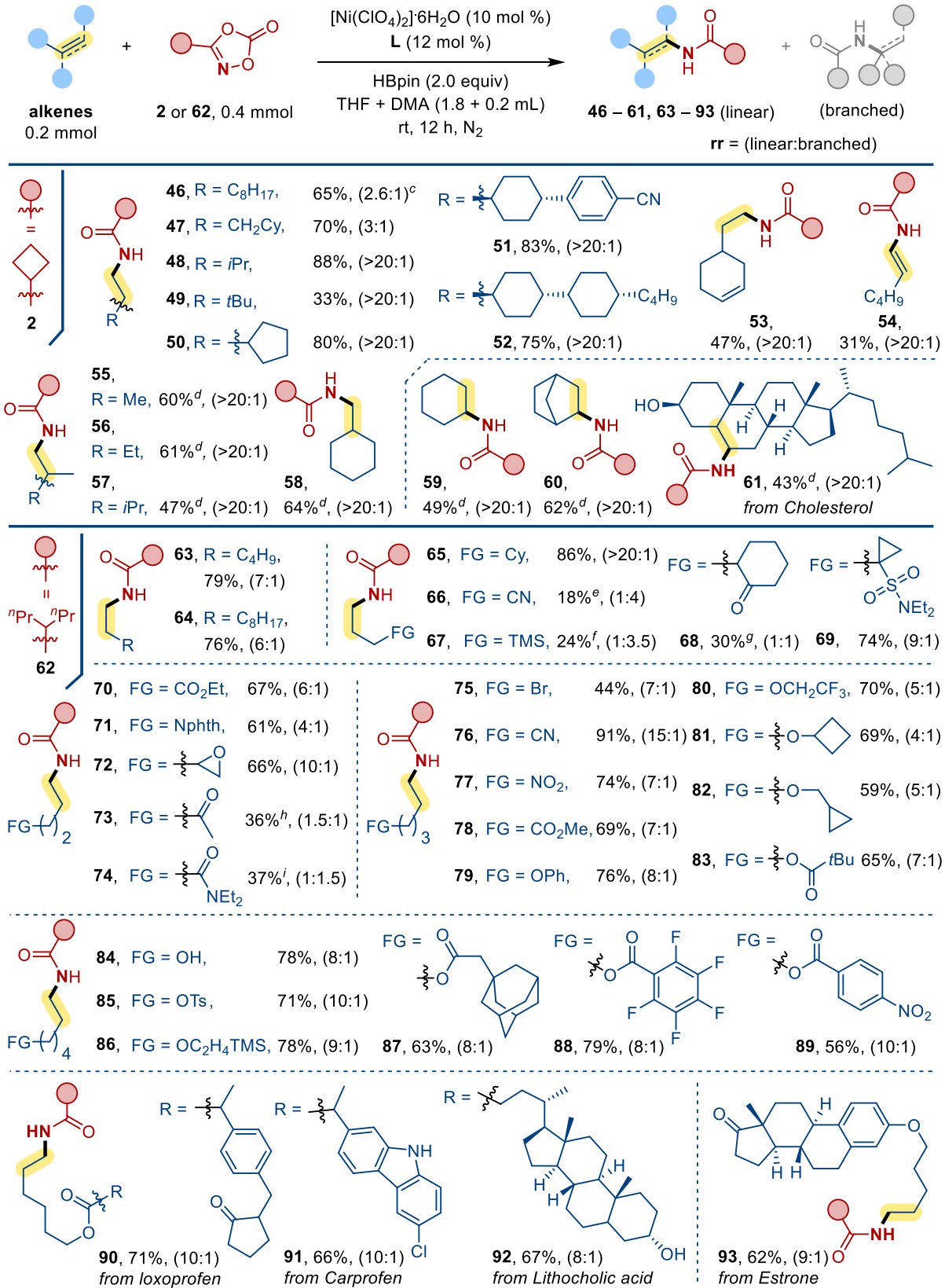

**Fig. 3 Scope of alkenes.** [a]Reaction conditions: alkenes (0.2 mmol), **2/62** (0.4 mmol), [Ni(ClO$_4$)$_2$]•6H$_2$O (10 mol %), **L4** (12 mol %), HBpin (2.0 equiv), THF + DMA (1.8 + 0.2 ml), in N$_2$ at room temperature for 12 h unless otherwise specified. [b]Isolated yield. [c]Regioisomeric ratio (**rr**) is defined as linear/branched. [d]**L3** instead of **L4**. [e]73% branched product. [f]70% branched product. [g]50% branched products. [h]53% branched products. [i]51% branched product.

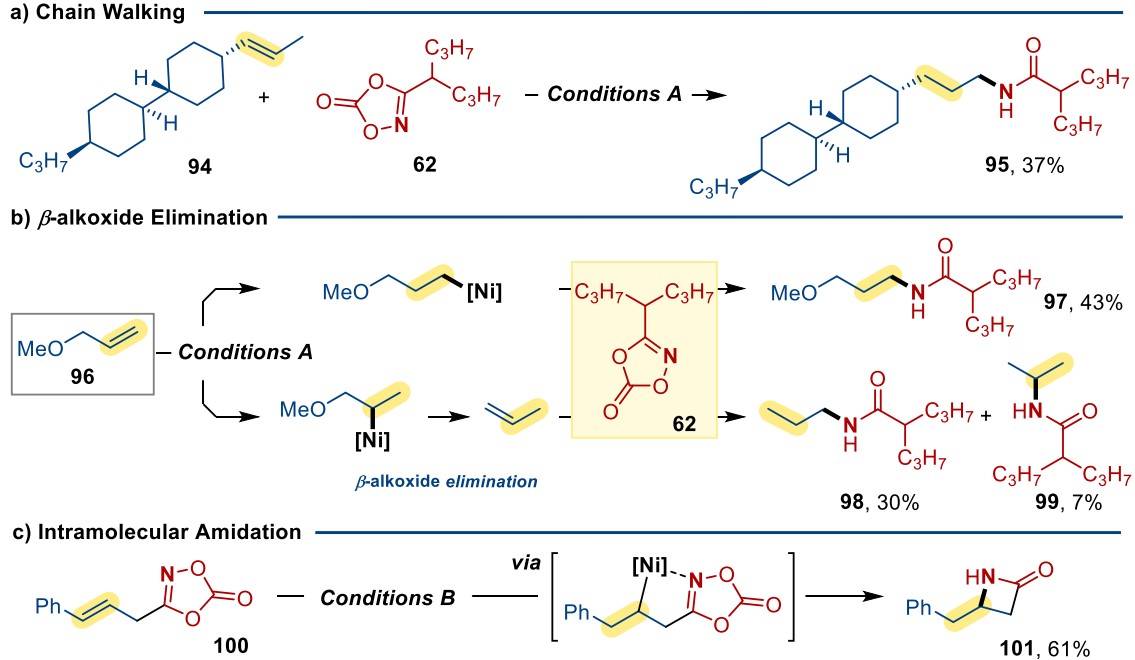

**Fig. 4 Examples involving chain walking, β-alkoxide elimination and intramolecular amidation. a** Chain walking. **b** β-alkoxide elimination.
**c** Intramolecular amidation. Conditions A: alkene (0.2 mmol), **62** (0.4 mmol), [Ni(ClO₄)₂]•6H₂O (10 mol %), **L4** (12 mol %), HBpin (2.0 equiv),
THF + DMA (1.8 + 0.2 ml), in N₂ at room temperature for 12 h. Conditions B: alkene (0.2 mmol), [Ni(ClO₄)₂]•6H₂O (10 mol %), **L4** (12 mol %), HBpin (2.0
equiv), MeCN + DCE (1.0 + 1.0 ml), in N₂ at room temperature for 12 h.

also examined. Substituents including hydroxyl, tosylate, 2-(trimethylsilyl)ethoxy and esters were found to be well tolerated: **84–89**, 56–79%. Effective transformations for those alkenes containing natural product motifs such as Loxoprofen (**90**: 71%), Carprofen (**91**: 66%), Lithocholic acid (**92**: 67%) and Estrone (**93**, 62%) were proved to be successful.

During the substrate scope study, we found that the reaction of a disubstituted alkene **94** afforded chain-walked product **95** in 37% yield (Fig. 4a). Moreover, the hydroamidation reaction of 3-methoxyprop-1-ene **96** with **62** gave the desired linear amide product **97** in 43% yield along with two unexpected products **98** (30%) and **99** (7%). The **98** and **99** formation can be accounted for by the prior β-elimination of the OMe group to generate prop-1-ene, which then reacted with **62** (Fig. 4b). The feasibility of the analogous intramolecular hydroamidation reaction was also demonstrated with 3-cinnamyl-1,4,2-dioxazol-5-one **100** as substrate. To our delight, the cyclized β-lactam 4-benzylazetidin-2-one **101** was formed in 61% yield implied that the potential coordination of dioxazolones (Fig. 4c).

To study the uniqueness and the working principle of dioxazolones as amidating reagents, hydroamidation with some common aminating and amidating reagents were tested (Fig. 5). Benzoic acid morpholin-4-yl ester **102** and 4-nitrotoluene **105** were found to be ineffective reagents in the current reaction system. While cyclohexyl isocyanate **108** could furnish the amide products with branched amides as the major product. Notably, comparable yield and selectivity was observed when benzoyl azide **110** was employed as the coupling partner instead of dioxazolones, which implies dioxazolone could also function as a nitrene source[27,28].

The reaction rate and selectivity has been probed by three sets of intermolecular competitive experiments. As depicted in Fig. 6a, upon treating an equimolar mixture of 3-cyclobutyl-1,4,2-dioxazol-5-one **2** and 3-ethyl-1,4,2-dioxazol-5-one **113** with **1** under the Ni-catalyzed conditions, the corresponding **3** and **5** were formed in 39% and 38%, respectively. This implies the reactivity of the dioxazolones bearing primary and secondary alkyl groups should be similar. Yet, the reaction with **2** and 3-(*tert*- butyl)-1,4,2-dioxazol-5-one **114** produced amide **3** predominantly in 69% vs. 9% for the **29** formation. Analogous crossover experiment using **1** and cyclohexene **115** with **2** as coupling partner was performed under the Ni-catalyzed conditions, amide **3** derived from the less hindered alkene **1** was formed in 75% yield vs. 23% yield for **59** formation. These findings support the steric sensitivity of the hydroamidation reactions.

A stepwise hydroboration/amidation pathway is improbable (Fig. 6b). Under the optimized conditions, the coupling reactions between **1** and **2** produced **3** in 88% yield. However, when the reaction was performed without the dioxazolone, ethylidenecyclohexane **116** (55%) was obtained exclusively without any hydroboration product **117** being formed (see Supporting Information)[29]. When subjecting a separately synthesized **117** to Ni-catalyzed conditions with dioxazolone **2**, no amide **3** was not obtained[30]. These results are inconsistent with a stepwise hydroboration/amidation pathway.

Deuterium labeling studies were also performed to study the reaction mechanism (Fig. 7a). The Ni-catalyzed coupling reaction of **1** and **2** was performed under the standard conditions with DBpin instead of HBpin. HNMR and HRMS analysis revealed two deuterium exchanged products (**122**: **123** ~ 7: 3) were formed. Plausibly, alkene insertion by [NiD] should generate two regioisomeric d-alkylnickel intermediates **118** (Markovnikov) and **119** (anti-Markovnikov) in a 3: 7 ratio[31–35]. While **119** should react directly with **2** to give **122**, the formation of **123** should proceed through reversible β-H elimination/insertion (via intermediate **120** and **121**)[36–43]. Similar results were obtained when **124** was treated with **125** under the Ni-catalyzed conditions (**126**: **127** ~ 7: 3). Based on these findings, we postulated the Curtin-Hammett selectivity to be proceeded by the reversible elimination/insertion of nickel-hydride complexes, followed by an irreversible C–N bond coupling[44]. To scrutinize this hypothesis, we investigated the regioselectivity of this reaction by comparing the

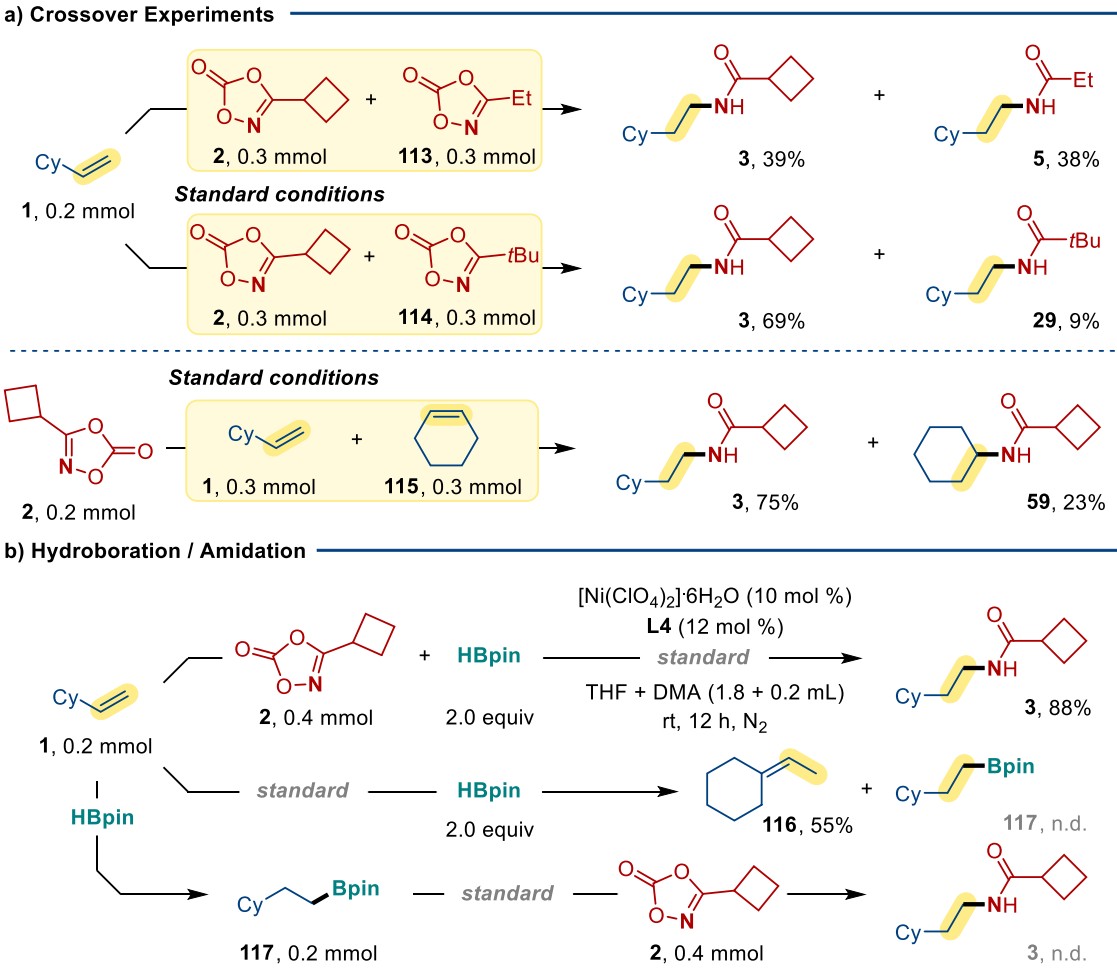

**Fig. 5 Hydroamination/hydroamidation with common aminating or amidating reagents.** Standard conditions: alkene (0.2 mmol), aminating (amidating) reagents (0.4 mmol), [Ni(ClO₄)₂]•6H₂O (10 mol %), **L4** (12 mol %), HBpin (2.0 equiv), THF + DMA (1.8 + 0.2 ml), in N₂ at room temperature for 12 h. All yields were presented as NMR yield.

**Fig. 6 Mechanistic investigation—competition experiments and control hydroboration/amidation reactions. a** Crossover experiments. **b** Hydroboration/amidation. Standard conditions: [Ni(ClO₄)₂]•6H₂O (10 mol %), **L4** (12 mol %), HBpin (2.0 equiv), THF + DMA (1.8 + 0.2 ml), in N₂ at room temperature for 12 h. The yields presented in part **a** are all isolated yield. The yield of **3** presented in part **b** is NMR yield. The yield of **116** presented in part **b** is GC yield.

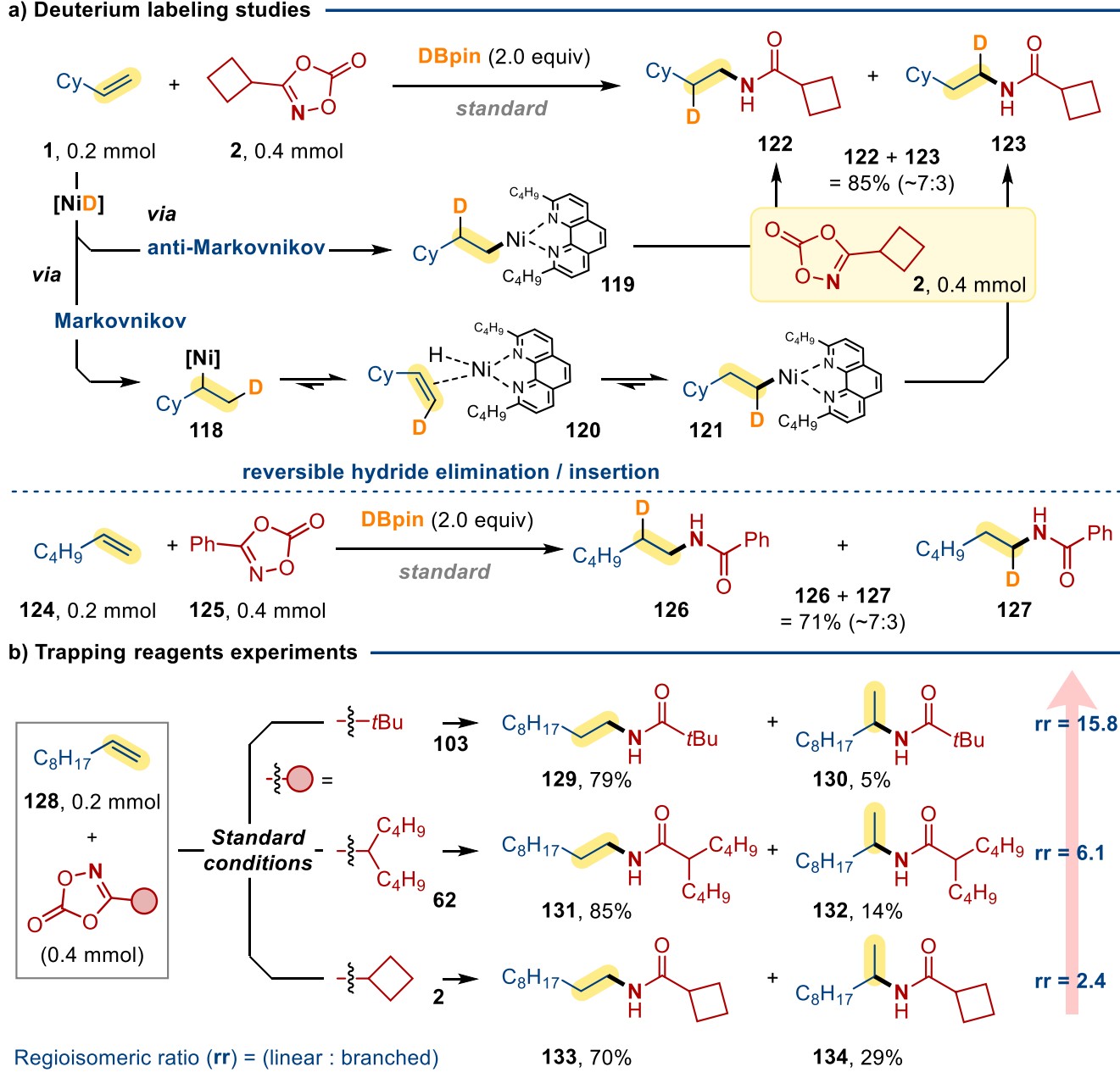

**Fig. 7 Mechanistic study—deuterium labeling studies and steric effect on regiosimeric ratio. a** Deuterium labeling studies. **b** Trapping reagents experiments. Standard conditions: [Ni(ClO$_4$)$_2$]·6H$_2$O (10 mol %), **L4** (12 mol %), HBpin/DBpin (2.0 equiv), THF + DMA (1.8 + 0.2 ml), in N$_2$ at room temperature for 12 h. All the yields presented are NMR yield.

results of three different dioxazolones (**2**, **62**, **103**) (Fig. 7b). Apparently, the linear/branched regioisomeric ratios (**rr** = 2.4, 6.1, 15.8) were very much dependent on the dioxazolone reagents (i.e., higher linear selectivity for bulkier dioxazolones). This finding supports the operation of Curtin-Hammett equilibrium prior to the irreversible C–N coupling step.

To gain further insight into the reaction mechanism and the origin of the anti-Markovnikov selectivity, we performed DFT calculations at the ωB97XD/SDD (Ni)/6-31G(d,p) level of theory (see Supporting Information for the computational details). Figure 8 depicts the energy profile calculated for the derived reaction mechanism. With [Ni(ClO$_4$)$_2$]·6H$_2$O as the precursor catalyst in the presence of propene, HBpin, and 2,9-disubsituted-1,10-phenanthroline as the ligand, a Ni(II) hydride intermediate (A) is formed as the active species of the reaction. Noted that the product molecule-coordinated Ni(II) hydride species G be the resting state according

to the energetics we obtained, we took G as the energy reference point in the energy profile. Ligand exchange of the model propene substrate for the model product molecule transforms the resting state G to the Ni(II) hydride active species A, followed by alkene insertion to form the agostic species B, which is further converted to the square planar complex C after coordination of dioxazolone. The release of CO$_2$ from the coordinated dioxazolone in the square planar Ni(II) complex C is rate-determining, with a see-saw shaped rate-determining transition state TSC–D, leading to a Ni(IV) complex D with an overall reaction barrier of 26.4 kcal mol$^{-1}$. Here, it should be pointed out that although Ni(IV) species are rare, organometallic Ni(IV) complexes containing alkyl and N ligands have been reported[45], and a few nickel-catalyzed reactions have proposed the intermediacy of Ni(IV)[46–48]. D is a four-coordinate d6 species. Such species are rare, but there are precedents with Ru(II), Rh(III) and Ir(III) centers[49–51]. HBpin acts as a Lewis acid and attaches to

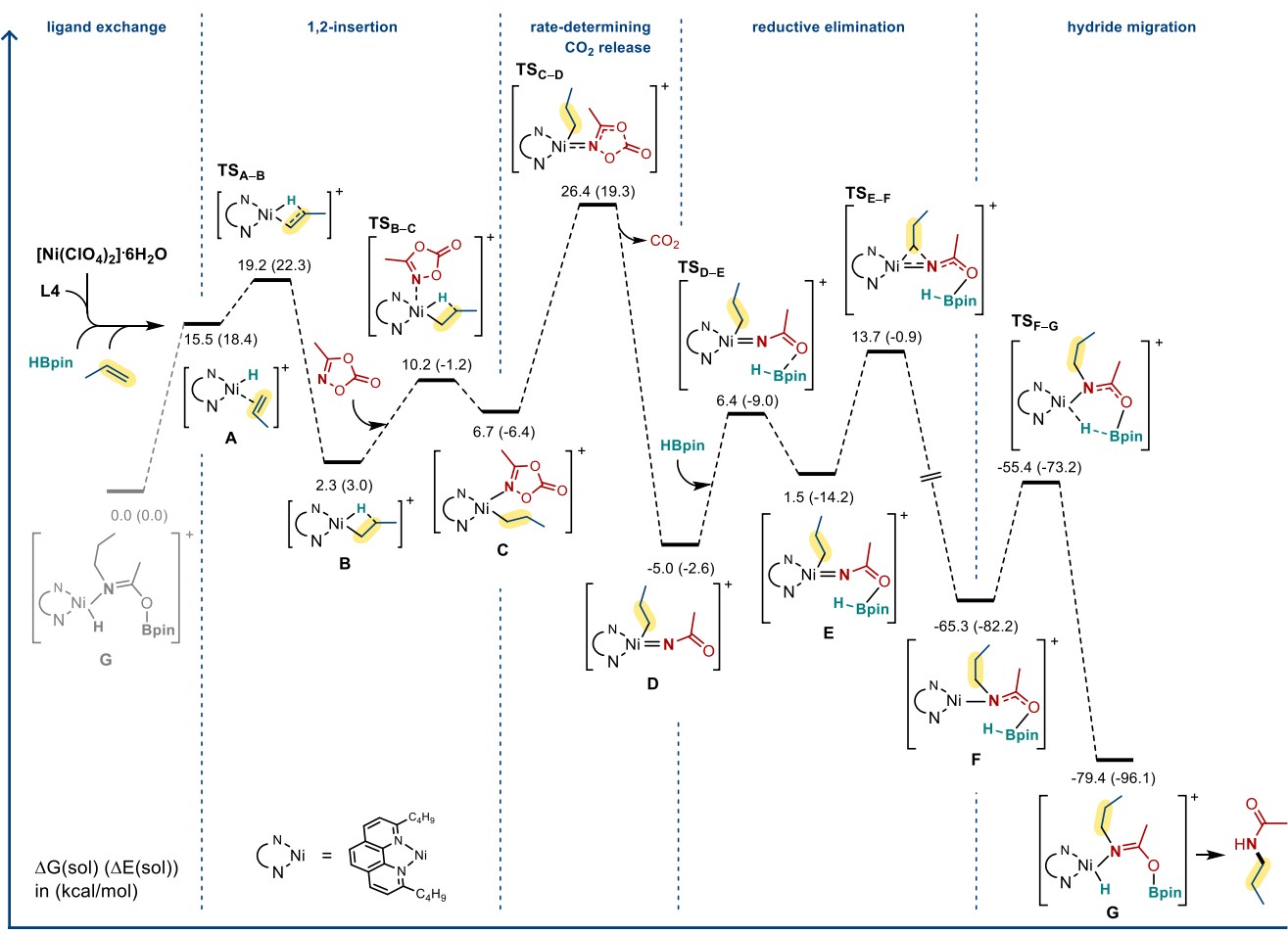

**Fig. 8 Gibbs free energy profile calculated for the Ni-catalyzed reaction of propene with 3-methyl-1,4,2-dioxazol-5-one.** The relative Gibbs energies and electronic energies (in parentheses) are given in kcal mol$^{-1}$.

the oxygen in D to form the intermediate E, from which a very facile reductive elimination gives Ni(II) complex F involving a drop of the free energy of the system significantly. Finally, the intermediate F undergoes intramolecular hydride migration from the attached HBpin to Ni to form the resting state G that contains a coordinated product molecule. Subsequent ligand exchange should regenerate the active species A.

One may query why the dioxazolone would not react with the NiH species (prior to olefin insertion), leading to nitrene insertion into the Ni–H bond. To address this question, we calculated the CO$_2$ extrusion transition state for the reaction of dioxazolone with the NiH species and found that this process is kinetically less favorable by 7.9 kcal mol$^{-1}$ than the corresponding one with the Ni-alkyl species shown in Fig. 8. We reasoned that an alkyl ligand should be relatively more electron-releasing than a hydride ligand, thereby facilitating the CO$_2$ extrusion process that raises the formal oxidation state of the nickel center. Based on this analysis, we concluded that alkene insertion to NiH should occur prior to the CO$_2$ extrusion of the dioxazolones. We also examined the possible Curtius rearrangement from the species **D**. The rearrangement barrier was calculated to be higher by 0.8 kcal mol$^{-1}$ than that calculated for **D → TS$_{E–F}$** in Fig. 8. The barrier difference is expected to increase significantly if the experimentally-used dioxazolone containing a cyclobutyl substituent is employed in the calculations. The steric bulky metal fragment likely suppresses the rearrangement process.

We also briefly examined the possible involvement of Ni(I)–H species. Our additional calculation results indicated that the Ni(I)–H

species shows extremely poor capability of coordinating the model alkene substrate. The alkene complex of Ni(I)–H was calculated to lie higher in free energy by 32.8 kcal mol$^{-1}$ than its resting state [the Ni(I) analogue of G] (see Supporting Information). This implies that our system is unlikely to involve Ni(I)–H species.

Regarding to the finding that different ligands display different linear-to-branch selectivity, DFT calculations were carried out on the resting state **G** and the rate-determining transition state **TS$_{C–D}$** with three different models: reaction of cyclohexylethene with 3-cyclobutyl-1,4,2-dioxazol-5-one employing the ligands 2-methyl-1,10-phenanthroline (**L2**), 2,9-dimethyl-1,10-phenanthroline (**L3**), and 2,9-di-n-butyl-1,10-phenanthroline (**L4**). Referring to Fig. 9, the calculated barrier differences between the Markovnikov hydroamidation transition state (branched) **TS-B$_{C–D}$** and the anti-Markovnikov hydroamidation transition state (linear) **TS-L$_{C–D}$** (ΔG(**TS-B$_{C–D}$**)−ΔG(**TS-L$_{C–D}$**)) are 0.3 kcal mol$^{-1}$ (**L2**), −0.5 kcal mol$^{-1}$ (**L3**) and −6.1 kcal mol$^{-1}$ (**L4**), which agree qualitatively well with the experimentally observed trend (for **L2**, 26%: 38%; for **L3**, 57%: 22%; for **L4**, 99%: n.d.); the calculated overall reaction barriers are 35.6 kcal mol$^{-1}$ (**L2**), 31.7 kcal mol$^{-1}$ (**L3**) and 30.9 kcal mol$^{-1}$ (**L4**) also agree qualitatively well with the experimentally observed trend in the reaction yields.

When the two substituents on the phenanthroline ligand were becoming bulkier as exemplified in **L2** to **L4**, the barrier difference becomes more negative (i.e., more anti-Markovnikov), and the calculated energy barrier decreases. This selectivity should be originated from the difference of steric hindrance at rate-determining transition state **TS$_{C–D}$**. In **TS-**

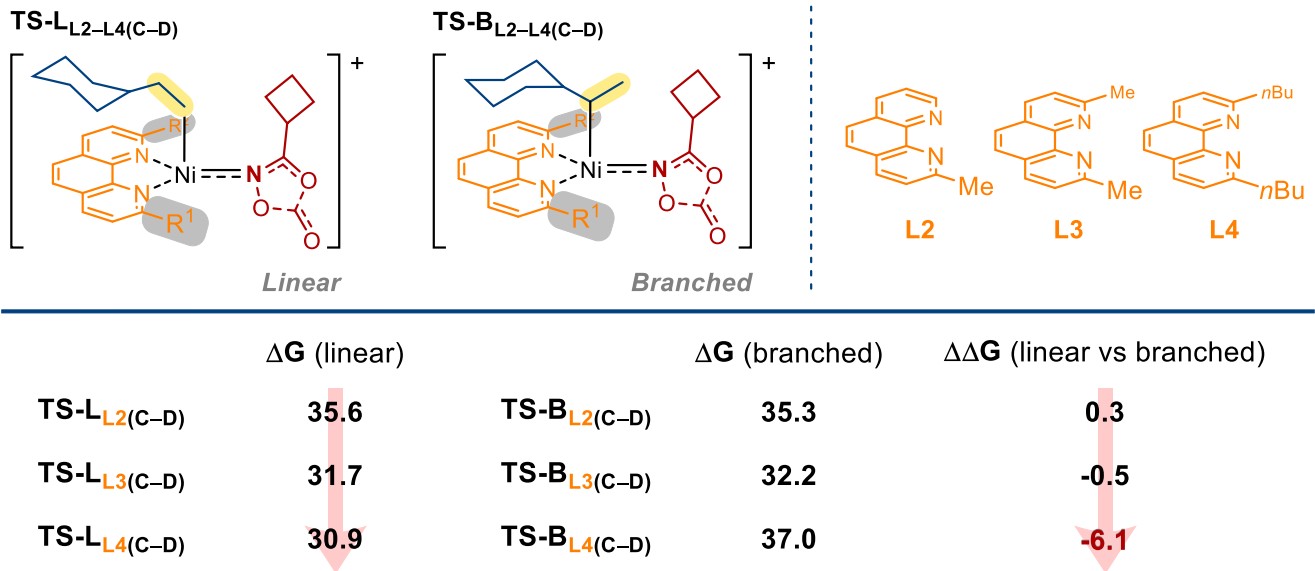

| | ΔG (linear) | | ΔG (branched) | ΔΔG (linear vs branched) |
|---|---|---|---|---|
| TS-L$_{L2(C-D)}$ | 35.6 | TS-B$_{L2(C-D)}$ | 35.3 | 0.3 |
| TS-L$_{L3(C-D)}$ | 31.7 | TS-B$_{L3(C-D)}$ | 32.2 | -0.5 |
| TS-L$_{L4(C-D)}$ | 30.9 | TS-B$_{L4(C-D)}$ | 37.0 | -6.1 |

**Fig. 9 Relative barriers calculation for the linear (anti-Markovnikov) hydroamidation vs. branched (Markovnikov) hydroamidation.** Rate-determining transition states **TS-B$_{L2-L4(C-D)}$** and **TS-L$_{L2-L4(C-D)}$** for Markovnikov hydroamidation (leading to branched product) and anti-Markovnikov hydroamidation (leading to linear product), respectively. The relative Gibbs free energies are given in kcal mol$^{-1}$.

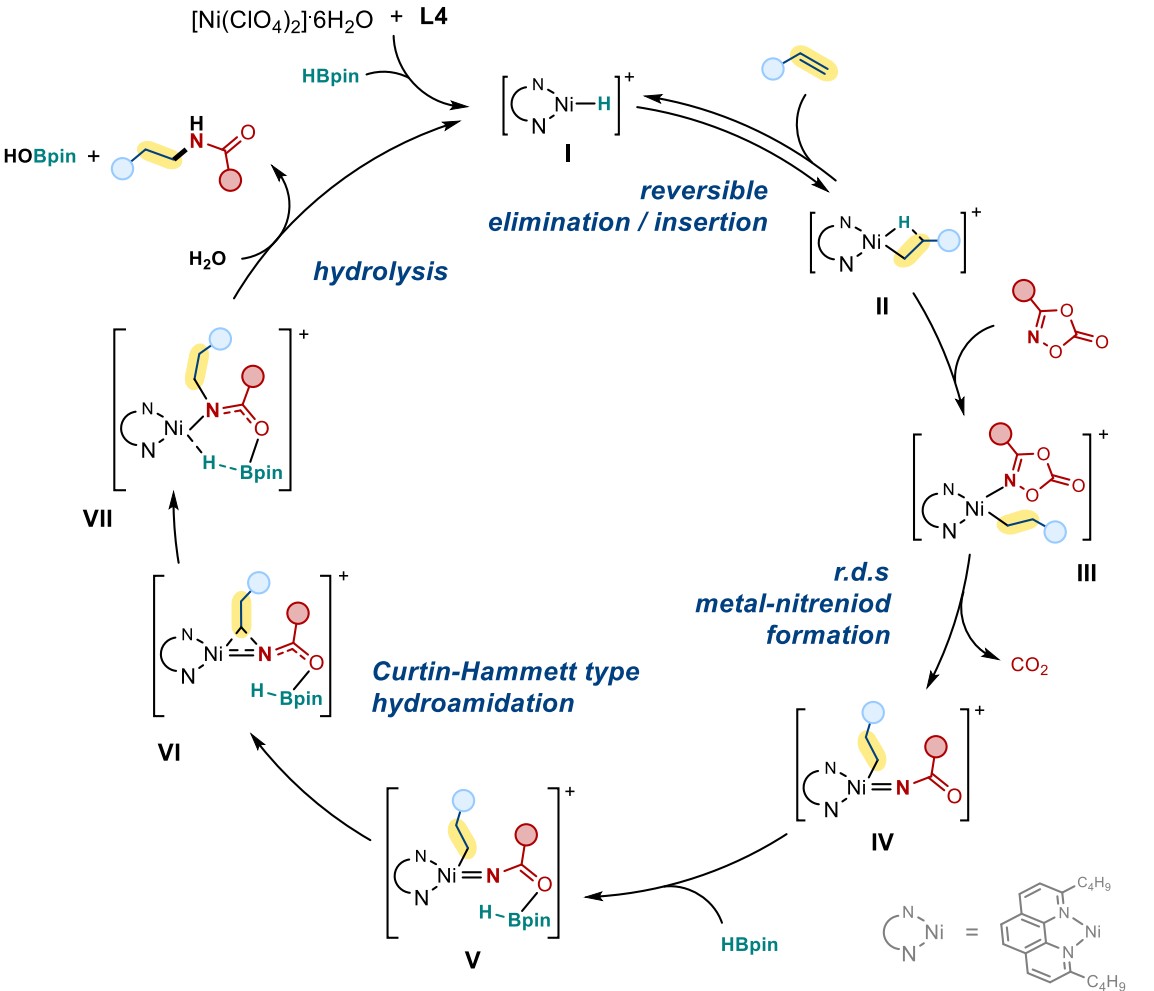

**Fig. 10 Proposed mechanism for the NiH-catalyzed ligand enabled Curtin-Hammett type hydroamidation.** Our proposed NiH-catalyzed ligand enabled Curtin-Hammett type hydroamidation mechanism.

$L_{C-D}$, the cyclohexyl group is located further from the two substituents on the phenanthroline ligand when compared with the situation in $TS-B_{C-D}$. In model L2, the steric effect becomes relatively unimportant. Then, the electronic effect dominates so that $TS-B_{L2C-D}$ becomes relatively more stable than $TS-L_{L2C-D}$ because the former has a stronger Ni–C bond, which contains a secondary alkyl ligand, whereas the latter contains a primary alkyl ligand. When the steric hindrance increases from L2 to L4, the steric effect becomes dominant, thus, the barrier differences change from positive to negative, causing a reversal of selectivity from Markovnikov to anti-Markovnikov. The energy barrier decreases from L2 to L4 should originate from the instability of the resting state G. G has a more sterically-hindered structure than $TS_{C-D}$. Therefore, when the reactants are bulkier, the energy difference between G and $TS_{C-D}$ decreases. Clearly, the DFT results show that the proposed mechanism explains the experimentally observed trends in selectivity and reactivity when changing the size of substituents on the phenanthroline ligands.

Based on the mechanistic investigations and DFT calculations, the proposed mechanism is depicted in Fig. 10. The active nickel hydride I is generated by treating $[Ni(ClO_4)_2] \cdot 6H_2O$ with HBpin and 2,9-dibutylphenathroline (L4). A reversible 1,2-insertion would then take place to give an alkylnickel intermediates II. The bulky ligand plays an important role to facilitate the anti-Markovnikov selectivity. Coordination of dioxazolone to the alkylnickel II generates complex III, and the subsequent rate-determining irreversible amidation gives the nickel-nitrenoid species IV. Coordination of an additional HBpin to the carbonyl group affords intermediate V and assists the following Curtin-Hammett type hydroamidation to generate intermediate VII via VI. A final hydrolysis process would regenerate the active nickel hydride I and furnish the linear N-alkyl amides.

## Conclusion

In conclusion, we developed a ligand-controlled Ni-catalyzed intermolecular hydroamidation of unactivated alkenes, and N-alkyl amides in predominantly anti-Markovnikov selectivity were obtained in excellent yields. This protocol is applicable to a wide range of substrate scope including some architecturally unique natural products. Since dioxazolones could be prepared from abundant carboxylic acid feedstocks, this hydroamidation protocol would be a powerful tool for easy conversion of hydrocarbon feedstocks to functionally and structurally diverse amides to cater for specific applications. Further control experiments and DFT calculations implied that the Curtin-Hammett selectivity model should be in operation.

## Methods

**General procedure for Ni-catalyzed cross-coupling reaction between alkenes and 1,4,2-dioxazol-5-ones**. In a nitrogen-filled glovebox, a 8 ml vial containing a magnetic stir bar was charged with $Ni(ClO_4)_2 \cdot 6H_2O$ (7.3 mg, 10 mol %) and 2,9-dibutyl-1,10-phenanthroline (7.0 mg, 12 mol %). Anhydrous THF (1.8 ml) and DMA (0.2 ml) were then added to the mixture via syringes. The vial was then screw-capped and stirred for 10 min at room temperature to give a brownish yellow [Ni + L] standard solution. Alkene (0.20 mmol, 1.0 equiv) and 1,4,2-dioxazol-5-one (0.40 mmol, 2.0 equiv) were added to a separate 8 ml vial with a magnetic stirrer bar. The [Ni + L] standard solution was transferred to the alkene-dioxazolone mixture with vigorous stirring. 4,4,5,5-Tetramethyl-1,3,2-dioxaborolane [HBpin, (0.40 mmol, 2.0 equiv)] was then added dropwise to the mixture via a syringe. A dark green solution mixture would appear upon addition of HBpin. The vial was then capped and removed from the glovebox. The mixture was then stirred at room temperature for 12 h. The crude mixture was transferred to a 25 ml round bottom flask and concentrated in vacuo. The residue was then filtered through a short-packed column with silica gel with ethyl acetate as eluent. The residue was again concentrated in vacuo. The residue was purified by column chromatography (n-hexane: ethyl acetate = 5: 1 to 1: 1), and the desired amide product can be visualized by TLC using $KMnO_4$ stain.

Supplementary Methods were provided in detail in the Supporting Information File.

## Data availability

All data are available from the corresponding authors upon reasonable request. Supplementary Information contains cif file for crystal structure, DFT calculation and the NMR spectra. The X-ray crystallographic coordinates for structures reported in this study have been deposited at the Cambridge Crystallographic Data Centre (CCDC), under deposition numbers 2052631 (40). The cif files for crystal structure is available in Supplementary Data 1. The DFT calculation is provided in Supplementary Data 2. The NMR spectra of all the compounds are available in Supplementary Data 3. It can be declared that all the relevant data are provided in the article and its Supplementary Information files. The data can be obtained free of charge from The Cambridge Crystallographic Data Centre via www.ccdc.cam.ac.uk/data_request/cif.

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

## Acknowledgements
The authors thank The Hong Kong Research Grants Council (153152/16P, 153023/17P, C5023-14G and HKUST16300021) for financial support. B.D and C.-M.C. are grateful for the Postdoctoral Fellowships generously supported by State Key Laboratory of Chemical Biology and Drug Discovery.

## Author contributions
B.D. and Y.O. performed the experiments. W.-Y.Y. and C.-M.C. directed the project. K.C. and Z.L. conducted the density functional theory calculations. All authors contributed to the preparation of the manuscript and the Supplementary Information.

## Competing interests
The authors declare no competing interests.

## Additional information

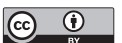

