## [Peer Review File · Communications Chemistry]

Reviewers' comments:

Reviewer #1 (Remarks to the Author):

Yu and co-workers reported a NiH catalyzed hydroamidation of aliphatic alkenes using 1,4,2-Dioxazol-5-ones as the amide source. The catalytic conditions are analogous to Chang's amidation of alkynes. However, the current system encompasses a broad range of mostly terminal alkenes and some internal alkenes. The selectivity with regard to the alkene structures and amide precursor are carefully studied. Direct amidation of aliphatic alkenes is particularly challenging. This paper certainly deserves publication.

Minor issues:

1. Figure 1b, the first equation, the intermediate should be an alkyl nickel instead of alkenyl nickel.
2. table 2, for compounds 41, 43 and 44, the stereochemistry should be indicated.
3. page 7, bottom right, change Ome to OMe.
4. table 3, for compounds 51 and 52, the relative configurations should be indicated.
5. figure 2, "exceptional examples" looks odd.
6. figure 5a, the Bu group is directly connected to nitrogen. This is a bit misleading.
7. page 15, bottom, change "insetion" to insertion.

Reviewer #2 (Remarks to the Author):

This manuscript by Lin and Yu describes the NiH-catalyzed hydroamidation of unactivated alkenes with HBpin as the hydride source and 1,4,2-dioxazol-5-ones as the nitrogen electrophile. High selectivity for the anti-Markovnikov product arises from the use of diBuphen as ligand and a combination of steric encumbrance on the alkene substrate and/or amine electrophile. Conceptually the work builds on related precedents in CuH and NiH catalysis, summarized in Figures 1a and 1b. Specifically, several directed and non-directed hydroamination methods under NiH catalysis have been recently described using various nitrogen electrophiles (Refs. 9, 10, 17–22). 1,4,2-Dioxazol-5-ones have been employed for hydroamidation of alkynes (Ref. 23) and alkenes containing thioether directing groups (Ref. 20), with the latter report coming from the same group as the present manuscript. Hence, the main advance in the present study is generalizing alkene hydroamidation with 1,4,2-dioxazol-5-ones to terminal alkenes lacking directing groups and achieving high anti-Markovnikov selectivity under conditions adapted from Ref. 20. Based on prior publications, it could be argued that this is an iterative advance with somewhat modest conceptual novelty. However, in my view, the development of this method it is a fairly significant accomplishment, and its reactivity and selectivity profile is distinct and complementary to related CuH methodology.

Regarding the method itself, the scope is amply demonstrated across upwards of 100 examples. Monosubstituted alkenes, 1,1-disubstituted alkenes, and cyclic internal alkenes are demonstrated, as is one example of a terminal alkyne. A series of competition experiments, isotope labeling experiments, and DFT computations are consistent with a mechanism involving reversible hydronickelation, turnover-limiting and regiodetermining oxidative addition, and finally reductive elimination. The proposed mechanism involves some interesting and not-immediately-obvious features, including a proposed Ni(IV)-nitrenoid and product-bound Ni(II)-H intermediate.

Based on these considerations, I support publication of this manuscript in Communications Chemistry following the minor revisions below:

(1) In Figure 1, panel b, top scheme involving the alkene: the intermediate and the product should not have the double bond still present

(2) Throughout all of the figures, the representation of the ligand is confusing, since the "short form" version of the ligand is itself a completely different chemical N,N-dibutyl-ethylene diamine. Note that in Figure 6, the ligand is drawn in a different way. The ligand can be consistently represented with a generic N-N structure with a curved line connecting the N atoms, and this will avoid confusion.

(3) In the introduction, I find it odd that the authors discuss a previous study by Chang on NiH-catalyzed alkyne hydroamidation as follows: "During preparation of this manuscript, Chang and co-workers demonstrated... (Ref. 23)." Chang's study was submitted in Jan 2021 and published in Apr 2021 (15 months ago), and in the meantime the authors themselves published another study on the use of 1,4,2-dioxazol-5-ones (Sept 2021) that is not discussed in this section. It seems this section needs to be updated considering the chronology of the published literature.

(4) What is the relative stereochemistry of product 25?

(5) Do the authors have an explanation for why the regioselectivity changes in the case of but-3-enenitrile and allyltrimethylsilane?

(6) For compound 52, only some of the relative stereochemistry is defined.

(7) In Figure 4a: this is not a "crossover experiment" in the traditional sense of the term. I would call this a competition experiment.

(8) In the footnotes for Figure 4, DBpin is mentioned, but I do not see it used in these experiments.

(9) Given that a Ni(II)-H species is proposed as the catalyst resting state, are the authors able to detect any evidence of a metal-hydride species by NMR?

(10) Though the manuscript is generally well written, there are several typos and grammatical/stylistic issues that can be corrected prior to publication:

- Abstract: "nitrogen function" -> "nitrogen-based functional group"
- Page 1: insert comma before "such as peptides"
- Page 2: "hydroxy amines" -> "hydroxy amine derivatives"
- Page 3: "hydroamidation for linear amides" -> "hydroamidation to access linear amides"
- Page 5: "we study" -> "we studied"
- Page 5: missing space in "including1-pyrenebutyric"
- Page 5: extra hyphen in "1-allyl-N,N-diethylcyclopropane-1-sulfonamide"
- Page 10: insert comma before "respectively"
- Page 15: "rate-determining" (with hyphen)

Reviewer #3 (Remarks to the Author):

The manuscript entitled "NiH-Catalyzed anti-Markovnikov Hydroamidation of Unactivated Alkenes with 1,4,2-Dioxazol-5-ones for the Direct Synthesis of N-Alkyl Amides" by Du et al. highlights a catalytic way of adding an amide moiety from unactivated alkenes. This work advances from the current methodology of making amides, formerly restricted by the need for a Directing group (ref 16 & 22) or by regioselectivity controlled by sterics (ref 15). The reaction manifold stems from prior JACS work

from 2013 and Angewandte chemie work from 2016 (ref 15 & 16), as well as recent work from CCS chem. and JACS 2020 & 2021 (ref 21, 22 & 23, respectively). The work, however, appears to be an expansion of the concepts and scope of Meng et al's work (ref 10) but altered to produce the desired anti-Markovnikov result without requiring the need for additives. It also appears to be largely similar to Lyu et al's work but with alkenes instead of alkynes (ref 23). The scope of the transformation appears to be extremely broad as several functionalities were tested and found to be successful. However, the scope does not include unsymmetrical cyclic alkenes. Recommend adding 2 examples to showcase the breadth of the scope. Detailed mechanistic studies support the proposed mechanism. However, the deuterium experiments should be presented better. An uncited procedure to produce a deuterated hydride source, DBpin is provided; and the extent of deuteration is not given, which implies 100% deuteration was assumed. Furthermore, no ²H NMR spectra were provided for any of the products. Instead, the assumption of 100% deuterated DBpin provided 100% deuterated (at a particular position). Moreover, the % D-incorporation is missing for 122 and 123. It should be noted that the chain walking experiment may be aided in the use of isotope-labeling alkene, which should be conducted. As for the SI, it is well presented; however, the range on several NMRs are presented inconsistently. At least 0 to 10 ppm should be shown for most spectra and some spectra, which are noted in additional comments below, are not phased appropriately. Also, there are several low intensity ¹³C-NMR signals. The intramolecular amidation experiment to produce 101 is done with a different solvent system. This should be corrected or explained as to why. In the crossover experiments, instead of just 115 with 1 it would be good to include in the SI cyclooctene as well as it introduces a less-strained internal alkene. Also, Figure 4A is reads as competition studies, not a crossover study. I suggest changing the title for Figure 4A. Furthermore, while for the most part, the references are accurate and in line with the work presented, references 1-3 are not substantive enough for the statement "It also constitutes an important motif in many functional materials and medicinal products." Reference 1 shows a single drug with an amide functionality that does not at least stress the importance of said functionality in the mechanism of action for its desired application. Reference 2, as far as could be seen, does not have any of the few drugs listed in it with amide functionalities. Lastly, reference 3 only has a single drug "maraviroc" with an amide functionality. I recommend going over these issues prior to publication. Overall, I do believe that this work merits publication in Nature Chemistry Communications and should be accepted after the above comments are addressed, as well as the additional comments below, some of which are indicative of quite a few issues of grammar and punctuation that are prevalent in this paper.

Additional Comments:

- 1) Page 2, paragraph 1, line 5 - "complexes" should be changed to "complex"
- 2) Page 3, paragraph 2, line 6 - "nickalacycle" should be changed to "nickelacycle"
- 3) Page 3, paragraph 2, line 11 - One cannot determine what would or would not be appealing to an industry. Change "Direct regioselective hydroamination of directing group-free alkene would be appealing for the synthetic industry." to "Direct regioselective hydroamination of directing group-free alkenes may be appealing to the synthetic industry."
- 4) Page 3, paragraph 4, line 4 - add a comma after "precursors" and after "Ni(COD)₂".
- 5) Page 3, paragraph 4 - change "Reaction with DCE gave a better result (3: 90%), while running the reactions in MeCN and DMA solvent afforded 3 in 55% and 56% yields respectively." to "The reaction with DCE gave a better result (3: 90%), while running the reaction in MeCN and in DMA afforded 3 in 55% and 56% yields, respectively."
- 6) Page 4, include THF solvent screen results in the main Manuscript
- 7) Page 5, entry 5 is suggested to be in 80% yield range, when it is not.
- 8) Page 5, paragraph 2, line 2 - "group" should be changed to "groups"
- 9) Page 5, paragraph 2, line 8 - add a comma after "Presumably"
- 10) Page 5, paragraph 2, last line - "70 % yields respectively" should be changed to "70% yields, respectively"

- 11) Page 5, paragraph 1, line 3 – “transformation, no” should be changed to “transformation; no”
- 12) Page 5, paragraph 4, line 4 – “relative” should be changed to “relatively” and “the attributed” to “attributed”
- 13) Page 5, paragraph 4, line 5 – add a comma before and after “such as 4-vinylcyclohex-1-ene”
- 14) Page 5, paragraph 4, line 6 – place discussion of 53 as its own sentence.
- 15) Page 7, Paragraph 1, line 2 – add a comma before the word “respectively”
- 16) Page 7 & page 5 – abstain from using the words “interestingly”, “surprisingly”, and “remarkable”
- 17) Page 7, Table 2 Caption – b denoted for isolated yields. None of the yields are tagged with b and yet the SI reports all of the yields as isolated. Correct the discrepancy.
- 18) Page 9, Table 3 Caption – same issue as 5). Correct the discrepancy.
- 19) Page 9, Figure 2 Caption – Isolated yields should not be set as reference b should if they are all isolated yields.
- 20) Page 3, paragraph 2 line 2 – “were tested” should be changed to “was tested”
- 21) Page 3, paragraph 2 line 5 – “was observed” should be changed to “were observed”
- 22) For all figure captions, letters proceeding footnotes should be superscripted.
- 23) For all figure captions, if the statement applies to the majority of the chart, remove the letter annotation. Only add letter annotation for exceptions. Example: “Fig. 2 Exceptional examples. a Standard reaction conditions: alkene (0.2 mmol), 62 (0.4 mmol), [Ni(ClO₄)₂] \cdot 6H₂O (10 mol %), L4 (12 mol %), HBpin (2.0 equiv), THF + DMA (1.8 + 0.2 mL), in N₂ at room temperature for 12 h unless otherwise specified. b Isolated yield. c Without 62, MeCN + DCE (1.0 + 1.0 mL).” should be “Fig. 2 Exceptional examples. Standard reaction conditions: alkene (0.2 mmol), 62 (0.4 mmol), [Ni(ClO₄)₂] \cdot 6H₂O (10 mol %), L4 (12 mol %), HBpin (2.0 equiv), THF + DMA (1.8 + 0.2 mL), in N₂ at room temperature for 12 h unless otherwise specified. Isolated yield. aWithout 62, MeCN + DCE (1.0 + 1.0 mL).” The superscript “c” in the figure should be then changed to “a”.
- 24) Page 10, Paragraph 2, line 4 – add a comma before the word “respectively”
- 25) Page 10, paragraph 1. You used the word “also”, discussing the probability of a nitrene when there has yet to be a discussion on the mechanism altogether.
- 26) Page 11, paragraph 1, line 8 – “these finding” should be “these findings” or “this finding”
- 27) Page 13, Figure 6. The amide product after hydrolysis is missing a group on the nitrogen atom.
- 28) Page 14, paragraph 2, line 1 – add comma before “leading”
- 29) Page 14, paragraph 2, line 7 – “occurs” should be changed to “occur”
- 30) Page 15, paragraph 2, line 1 – “displaying” should be changed to “display”
- 31) Page 15, paragraph 3, line 9 – “hinderance” should be changed to “hindrance” and “, thus the” should be changed to “; thus, the”
- 32) Page 15, paragraph 4, line 3 – “1,2-insetion” should be changed to “1,2-insertion”
- 33) Page 16, paragraph 1, line 3 – “assist” should be changed to “assists”
- 34) Fix all mentions of “rate determing” to “rate-determining”
- 35) Fix all capitalization of Markovnikov. “anti-markovnikov” should be changed to “anti-Markovnikov”
- 36) Figure 2b – “beta elimination” should be changed to “beta-alkoxide” elimination to avoid confusion.
- 37) SI, Page 8. S17 looks squished.
- 38) SI – Fix phasing issues of ¹H NMR spectra S7, S21

Point by Point Response to Reviewers and Editorial Office

Reviewer #1 (Remarks to the Author):

Yu and co-workers reported a NiH catalyzed hydroamidation of aliphatic alkenes using 1,4,2-Dioxazol-5-ones as the amide source. The catalytic conditions are analogous to Chang's amidation of alkynes. However, the current system encompasses a broad range of mostly terminal alkenes and some internal alkenes. The selectivity with regard to the alkene structures and amide precursor are carefully studied. Direct amidation of aliphatic alkenes is particularly challenging. This paper certainly deserves publication.

Minor issues:

1. Figure 1b, the first equation, the intermediate should be an alkyl nickel instead of alkenyl nickel.
> *Thank you for this kindly advise, that have been revised.*
2. table 2, for compounds 41, 43 and 44, the stereochemistry should be indicated.
> *The corresponding stereochemistry has been added to manuscripts.*
3. page 7, bottom right, change Ome to OMe.
> *Thank you for this kindly advise, that have been revised.*
4. table 3, for compounds 51 and 52, the relative configurations should be indicated.
> *Thanks for this advice, those relative configurations have been revised.*
5. figure 2, "exceptional examples" looks odd.
> *The description has been revised to 'Examples involving chain walking, B-elimination, and intramolecular amidation'.*
6. figure 5a, the Bu group is directly connected to nitrogen. This is a bit misleading.
> *The nBu notation has been revised to C4-H9 for better presentation.*
7. page 15, bottom, change "insetion" to insertion.
> *Thank you for this kindly advise, that have been revised.*

Reviewer #2 (Remarks to the Author):

This manuscript by Lin and Yu describes the NiH-catalyzed hydroamidation of unactivated alkenes with HBpin as the hydride source and 1,4,2-dioxazol-5-ones as the nitrogen electrophile. High selectivity for the anti-Markovnikov product arises from the use of diBuphen as ligand and a combination of steric encumbrance on the alkene substrate and/or amine electrophile. Conceptually the work builds on related precedents in CuH and NiH catalysis, summarized in Figures 1a and 1b. Specifically, several directed and non-directed hydroamination methods under NiH catalysis have been recently described using various nitrogen electrophiles (Refs. 9, 10, 17–22). 1,4,2-Dioxazol-5-ones have been employed for hydroamidation of alkynes (Ref. 23) and alkenes containing thioether directing groups (Ref. 20), with the latter report coming from the same group as the present manuscript. Hence, the main advance in the present study is generalizing alkene hydroamidation with 1,4,2-dioxazol-5-ones to terminal alkenes lacking directing groups and achieving high anti-Markovnikov selectivity under conditions adapted from Ref. 20. Based on prior publications, it could be argued that this is an iterative advance with somewhat modest conceptual novelty. However, in my view, the development of this method it is a fairly significant accomplishment, and its reactivity and selectivity profile is distinct and complementary to related CuH methodology.

Regarding the method itself, the scope is amply demonstrated across upwards of 100 examples. Monosubstituted alkenes, 1,1-disubstituted alkenes, and cyclic internal alkenes are demonstrated, as is one example of a terminal alkyne. A series of competition experiments, isotope labeling experiments, and DFT computations are consistent with a mechanism involving reversible hydronickelation, turnover-limiting and regiodetermining oxidative addition, and finally reductive elimination. The proposed mechanism involves some interesting and not-immediately-obvious features, including a proposed Ni(IV)–nitrenoid and product-bound Ni(II)–H intermediate.

Based on these considerations, I support publication of this manuscript in Communications Chemistry following the minor revisions below:

1. In Figure 1, panel b, the top scheme involving the alkene: the intermediate and the product should not have the double bond still present

> *Thank you for this kindly advise, that have been revised.*

2. Throughout all of the figures, the representation of the ligand is confusing, since the “short form” version of the ligand is itself a completely different chemical N,N-dibutyl-ethylene diamine. Note that in Figure 6, the ligand is drawn in a different way. The ligand can be consistently represented with a generic N–N structure with a curved line connecting the N atoms, and this will avoid confusion.

> *Figures, Tables and Schemes regarding the notation nBu or nPr have been revised, (Fig 1, Table 1, Fig 2, Fig 5, Fig 6, and Fig 8).*

3. In the introduction, I find it odd that the authors discuss a previous study by Chang on NiH-catalyzed alkyne hydroamidation as follows: “During preparation of this manuscript, Chang and co-workers demonstrated… (Ref. 23).” Chang’s study was submitted in Jan 2021 and published in Apr 2021 (15 months ago), and in the meantime the authors themselves published another study on the use of 1,4,2-dioxazol-5-ones (Sept 2021) that is not discussed in this section. It seems this section needs to be updated considering the chronology of the published literature.

> *Thanks for the reviewer’s advice, our group’s work has been discussed in the introduction part which would make the chronology more suitable.*

4. What is the relative stereochemistry of product 25?

> *Product from (+/-)-4-Methylcyclohexanecarboxylic Acid.*

5. Do the authors have an explanation for why the regioselectivity changes in the case of but-3-enitrile and allyltrimethylsilane?

> *For the but-3-enitrile:*

One explanation: The π bond in cyano could form the π - π stacking with the phenanthroline around nickel central which is in the form of suitable cycle ring tension.

Another explanation: The branched carbon of the but-3-enitrile show more negative electricity which prefers to locate the [Ni] that makes the amidation occurs in the branched position.

For the allyltrimethylsilane:

We prefer the potential p-d π bond conjugation between the “Si” and the phenanthroline.

6. For compound 52, only some of the relative stereochemistry is defined.

> *Thank you for this kindly advise, that have been revised.*

7. In Figure 4a: this is not a “crossover experiment” in the traditional sense of the term. I would call this a competition experiment.

> *Thanks for the recommendation, the footnotes have been revised.*

8. In the footnotes for Figure 4, DBpin is mentioned, but I do not see it used in these experiments.

> *DBpin has been deleted in the footnotes.*

9. Given that a Ni(II)–H species is proposed as the catalyst resting state, are the authors able to detect any evidence of a metal–hydride species by NMR?

> *We have produced a series of experiments to gain the Mass or NMR data of the related Ni(II)-H complex but failed. We attribute it to the high activity of the species.*

10. Though the manuscript is generally well written, there are several typos and grammatical/stylistic issues that can be corrected prior to publication:

- Abstract: “nitrogen function” -> “nitrogen-based functional group”
- Page 1: insert comma before “such as peptides”
- Page 2: “hydroxy amines” -> “hydroxy amine derivatives”
- Page 3: “hydroamidation for linear amides” -> “hydroamidation to access linear amides”
- Page 5: “we study” -> “we studied”
- Page 5: missing space in “including1-pyrenebutyric”
- Page 5: extra hyphen in “1-allyl-N,N-diethylcyclopropane-1-sulfonamide”
- Page 10: insert comma before ‘respectively’
- Page 15: “rate-determining” (with hyphen)

> *Thanks for the kind reminder, these errors have been corrected.*

Reviewer #3 (Remarks to the Author):

The manuscript entitled “NiH-Catalyzed anti-Markovnikov Hydroamidation of Unactivated Alkenes with 1,4,2-Dioxazol-5-ones for the Direct Synthesis of N-Alkyl Amides” by Du et al. highlights a catalytic way of adding an amide moiety from unactivated alkenes. This work advances from the current methodology of making amides, formerly restricted by the need for a Directing group (ref 16 & 22) or by regioselectivity controlled by sterics (ref 15). The reaction manifold stems from prior JACS work from 2013 and Angewandte chemie work from 2016 (ref 15 & 16), as well as recent work from CCS chem. and JACS 2020 & 2021 (ref 21, 22 & 23, respectively). The work, however, appears to be an expansion of the concepts and scope of Meng et al's work (ref 10) but altered to produce the desired anti-Markovnikov result without requiring the need for additives. It also appears to be largely similar to Lyu et al's work but with alkenes instead of alkynes (ref 23). The scope of the transformation appears to be extremely broad as several functionalities were tested and found to be successful. However, the scope does not include unsymmetrical cyclic alkenes. Recommend adding 2 examples to showcase the breadth of the scope.

> *Thanks so much for this advice, we have tried several examples of unsymmetrical cyclic alkenes but without very good results. One successful example is the scope of 61 which has been presented in table 3.*

Detailed mechanistic studies support the proposed mechanism. However, the deuterium experiments should be presented better. An uncited procedure to produce a deuterated hydride source, DBpin is provided; and the extent of deuteration is not given, which implies

100% deuteration was assumed.

> Thanks for the reviewer's advice, more details experiments and the corresponding cited paper (Yu, H. C.; Islam, S. M.; Mankad, N. P. ACS Catal. 2020, 10, 3670-3675) have been added in supporting information.

Furthermore, no ²H NMR spectra were provided for any of the products. Instead, the assumption of 100% deuterated DBpin provided 100% deuterated (at a particular position). Moreover, the % D-incorporation is missing for 122 and 123.

> For the ²H NMR question: Thanks for this advice, one ²H NMR spectra of 35 has been added in supporting information.

> For the % D-incorporation question: The ¹H NMR of 122 with 123 shows a deviation in peak integration at δ 1.37 (t, J = 7.3 Hz, 1.32H) and δ 3.25 (t, J = 6.6 Hz, 1.68H) compared to non-deuterated products, resulting from deuterium incorporation at the α and β position. Based on the value of the integration, 122 and 123 is 100% deuterated without non-deuterated products. So, we calculated it to be approximately 7 : 3 ratio of 122 to 123 with 85% total yield. More detail has been given in supporting information.

It should be noted that the chain walking experiment may be aided in the use of isotope-labeling alkene, which should be conducted.

> Thanks so much for this academic advice, some experiments have been conducted in our lab. However, we failed in prepared the D-labelled alkene.

As for the SI, it is well presented; however, the range on several NMRs are presented inconsistently. At least 0 to 10 ppm should be shown for most spectra and some spectra, which are noted in additional comments below, are not phased appropriately. Also, there are several low intensity ¹³C-NMR signals.

> Thanks for the advice, parts of NMRs spectra have been presented properly for easily check-in supporting information.

The intramolecular amidation experiment to produce 101 is done with a different solvent system. This should be corrected or explained as to why.

> Thanks, actually it's reported without mistake. The use of the different solvent systems is based on our series of optimization. We want to give a higher yield of the reaction, so we selected present different solvent systems.

In the crossover experiments, instead of just 115 with 1 it would be good to include in the SI cyclooctene as well as it introduces a less-strained internal alkene.

> Thanks, as results showed in Fig.4a, the ratio of the 3 to 59 is nearly 3:1 which indicates the steric sensitivity of the hydroamidation reactions. Cyclooctene as a more sterically hindered alkene than vinylcyclohexane should give a similar result.

Also, Figure 4A is reads as competition studies, not a crossover study. I suggest changing the title for Figure 4A.

> Thank you for this kindly advise, that have been revised.

Furthermore, while for the most part, the references are accurate and in line with the work presented, references 1-3 are not substantive enough for the statement "It also constitutes an important motif in many functional materials and medicinal products." Reference 1 shows a single drug with an amide functionality that does not at least stress the importance of said functionality in the mechanism of action for its desired application. Reference 2, as far as could be seen, does not have any of the few drugs listed in it with amide functionalities. Lastly,

reference 3 only has a single drug "maraviroc" with an amide functionality. I recommend going over these issues prior to publication.

Overall, I do believe that this work merits publication in Nature Chemistry Communications and should be accepted after the above comments are addressed, as well as the additional comments below, some of which are indicative of quite a few issues of grammar and punctuation that are prevalent in this paper.

> Thanks, two more relativity references have been added to the manuscripts.

Additional Comments:

- 1) Page 2, paragraph 1, line 5 – “complexes” should be changed to “complex”
- 2) Page 3, paragraph 2, line 6 – “nickalacycle” should be changed to “nickelacycle”
- 3) Page 3, paragraph 2, line 11 – One cannot determine what would or would not be appealing to an industry. Change “Direct regioselective hydroamination of directing group-free alkene would be appealing for the synthetic industry.” to “Direct regioselective hydroamination of directing group-free alkenes may be appealing to the synthetic industry.”
- 4) Page 3, paragraph 4, line 4 – add a comma after “precursors” and after “Ni(COD)₂”.
- 5) Page 3, paragraph 4 – change “Reaction with DCE gave a better result (3: 90%), while running the reactions in MeCN and DMA solvent afforded 3 in 55% and 56% yields respectively.” to “The reaction with DCE gave a better result (3: 90%), while running the reaction in MeCN and in DMA afforded 3 in 55% and 56% yields, respectively.”
- 6) Page 4, include THF solvent screen results in the main Manuscript
> Thanks, entry 1 in Table 1 is the THF solvent screen results.
- 7) Page 5, entry 5 is suggested to be in 80% yield range, when it is not.
> Thanks, when heterogeneous atoms (such as “O”, “N”) in substrates the corresponding yields would be lower than 80%.
- 8) Page 5, paragraph 2, line 2 – “group” should be changed to “groups”
- 9) Page 5, paragraph 2, line 8 – add a comma after “Presumably”
- 10) Page 5, paragraph 2, last line – “70 % yields respectively” should be changed to “70% yields, respectively”
- 11) Page 5, paragraph 1, line 3 – “transformation, no” should be changed to “transformation; no”
- 12) Page 5, paragraph 4, line 4 – “relative” should be changed to “relatively” and “the attributed” to “attributed”
- 13) Page 5, paragraph 4, line 5 – add a comma before and after “such as 4-vinylcyclohex-1-ene”
- 14) Page 5, paragraph 4, line 6 – place discussion of 53 as its own sentence.
- 15) Page 7, Paragraph 1, line 2 – add a comma before the word “respectively”
- 16) Page 7 & page 5 – abstain from using the words “interestingly”, “surprisingly”, and “remarkable”
- 17) Page 7, Table 2 Caption – b denoted for isolated yields. None of the yields are tagged with b and yet the SI reports all of the yields as isolated. Correct the discrepancy.
- 18) Page 9, Table 3 Caption – same issue as 5). Correct the discrepancy.
- 19) Page 9, Figure 2 Caption – Isolated yields should not be set as reference b should if they are all isolated yields.

- 20) Page 3, paragraph 2 line 2 – “were tested” should be changed to “was tested”
- 21) Page 3, paragraph 2 line 5 – “was observed” should be changed to “were observed”
- 22) For all figure captions, letters preceding footnotes should be superscripted.
- 23) For all figure captions, if the statement applies to the majority of the chart, remove the letter annotation. Only add letter annotation for exceptions. Example: “Fig. 2 Exceptional examples. a Standard reaction conditions: alkene (0.2 mmol), 62 (0.4 mmol), [Ni(ClO₄)₂] · 6H₂O (10 mol %), L4 (12 mol %), HBpin (2.0 equiv), THF + DMA (1.8 + 0.2 mL), in N₂ at room temperature for 12 h unless otherwise specified. b Isolated yield. c Without 62, MeCN + DCE (1.0 + 1.0 mL).” should be “Fig. 2 Exceptional examples. Standard reaction conditions: alkene (0.2 mmol), 62 (0.4 mmol), [Ni(ClO₄)₂] · 6H₂O (10 mol %), L4 (12 mol %), HBpin (2.0 equiv), THF + DMA (1.8 + 0.2 mL), in N₂ at room temperature for 12 h unless otherwise specified. Isolated yield. aWithout 62, MeCN + DCE (1.0 + 1.0 mL).” The superscript “c” in the figure should be then changed to “a”.
- 24) Page 10, Paragraph 2, line 4 – add a comma before the word “respectively”
- 25) Page 10, paragraph 1. You used the word “also”, discussing the probability of a nitrene when there has yet to be a discussion on the mechanism altogether.
- 26) Page 11, paragraph 1, line 8 – “these finding” should be “these findings” or “this finding”
- 27) Page 13, Figure 6. The amide product after hydrolysis is missing a group on the nitrogen atom.
- 28) Page 14, paragraph 2, line 1 – add comma before “leading”
- 29) Page 14, paragraph 2, line 7 – “occurs” should be changed to “occur”
- 30) Page 15, paragraph 2, line 1 – “displaying” should be changed to “display”
- 31) Page 15, paragraph 3, line 9 – “hinderance” should be changed to “hindrance” and “, thus the” should be changed to “; thus, the”
- 32) Page 15, paragraph 4, line 3 – “1,2-insetion” should be changed to “1,2-insertion”
- 33) Page 16, paragraph 1, line 3 – “assist” should be changed to “assists”
- 34) Fix all mentions of “rate determining” to “rate-determining”
- 35) Fix all capitalization of Markovnikov. “anti-markovnikov” should be changed to “anti-Markovnikov”
- 36) Figure 2b – “beta elimination” should be changed to “beta-alkoxide” elimination to avoid confusion.
- 37) SI, Page 8. S17 looks squished.
- 38) SI – Fix phasing issues of ¹H NMR spectra S7, S21
> *Thanks for the kind reminder, these errors have been corrected.*

REVIEWERS' COMMENTS:

Reviewer #2 (Remarks to the Author):

The authors have thoroughly and thoughtfully revised the manuscript based on feedback from the previous round of peer review, and the manuscript now appears to be ready for publication as-is.

Reviewer #3 (Remarks to the Author):

The authors have addressed all comments and the manuscript is ready for publication. However, this is one minor issue with their SI. Their NMR spectrum range is 0-8 ppm for most cases. As highlighted in ACS JOC Supporting NMR standards (https://publish.acs.org/publish/author_guidelines?coden=jocceah), the recommended range is 0 to 10 ppm. While Nature Communication Chem is different from ACS, these standards are upheld by many journals. I leave it up to the editors to decide if the authors should amend this issue. However, I do recommend fixing all the NMR spectrum that to fit the viewing range of 0-10 ppm before publication.